# The Role of Gut Dysbiosis in the Loss of Intestinal Immune Cell Functions and Viral Pathogenesis

**DOI:** 10.3390/microorganisms11071849

**Published:** 2023-07-21

**Authors:** Farzaneh Fakharian, Siva Thirugnanam, David A. Welsh, Woong-Ki Kim, Jay Rappaport, Kyle Bittinger, Namita Rout

**Affiliations:** 1Department of Microbiology, Faculty of Biological Sciences and Technology, Shahid Beheshti University, Tehran 1983969411, Iran; 2Tulane National Primate Research Center, Covington, LA 70433, USA; 3Department of Microbiology and Immunology, Tulane University School of Medicine, New Orleans, LA 70112, USA; 4Department of Microbiology, Immunology and Parasitology, Louisiana State University School of Medicine, New Orleans, LA 70806, USA; 5Division of Gastroenterology, Hepatology and Nutrition, Children’s Hospital of Philadelphia, Philadelphia, PA 19104, USA; 6Tulane Center for Aging, Tulane University School of Medicine, New Orleans, LA 70112, USA

**Keywords:** gut dysbiosis, viral pathogenesis, γδ T cell, probiotics, FMT

## Abstract

The gut microbiome plays a critical role in maintaining overall health and immune function. However, dysbiosis, an imbalance in microbiome composition, can have profound effects on various aspects of human health, including susceptibility to viral infections. Despite numerous studies investigating the influence of viral infections on gut microbiome, the impact of gut dysbiosis on viral infection and pathogenesis remains relatively understudied. The clinical variability observed in SARS-CoV-2 and seasonal influenza infections, and the presence of natural HIV suppressors, suggests that host-intrinsic factors, including the gut microbiome, may contribute to viral pathogenesis. The gut microbiome has been shown to influence the host immune system by regulating intestinal homeostasis through interactions with immune cells. This review aims to enhance our understanding of how viral infections perturb the gut microbiome and mucosal immune cells, affecting host susceptibility and response to viral infections. Specifically, we focus on exploring the interactions between gamma delta (γδ) T cells and gut microbes in the context of inflammatory viral pathogenesis and examine studies highlighting the role of the gut microbiome in viral disease outcomes. Furthermore, we discuss emerging evidence and potential future directions for microbiome modulation therapy in the context of viral pathogenesis.

## 1. Introduction

The human gastrointestinal (GI) tract hosts an extensive and diverse population of microorganisms, including bacteria, viruses, protozoa, and fungi. Together, they form the complex and dynamic ecosystem known as the gut microbiota [1]. Notably, the gut microbiota performs vital functions across various physiological processes, influencing aspects of human health and well-being. These include the regulation of immune responses, maintenance of gut barrier integrity, metabolism, immunological function, and the proper functioning of the enteric nervous system [2,3,4,5]. Imbalances in the composition of the gut microbiota, referred to as gut dysbiosis, have been implicated in the development of several diseases, including autoimmune disorders, chronic inflammatory conditions, metabolic diseases, viral pathogenesis, obesity, type 1 and type 2 diabetes, autism-related conditions, and certain gastrointestinal cancers [3,6,7,8,9].

The interaction between the gut microbiota and intestinal mucosal immunity is complex, exerting manifold effects on homeostasis and disease processes [10]. Both the small and large intestines play crucial roles in the multifaceted interaction between the gut microbiota and intestinal mucosal immunity; while the small intestine is primarily responsible for nutrient absorption, the large intestine plays a key role as the primary site of fermentation and SCFA production [11]. The GI tract is the primary interface between the microbiota and the host immune system. The innate immune system provides the initial physical barrier against pathogens, while the adaptive immune system targets pathogens specifically and generates immunological memory [12]. The detection of pathogen-associated molecular patterns (PAMPs) by antigen-presenting cells (APCs) through pattern recognition receptors (PRRs) is integral to the innate immune response. PRRs enable APCs to identify PAMPs, which in turn initiates a cascade of inflammatory responses. The stimulation of naive T cells by PAMP-matured APCs, such as DCs, leads to the generation of distinct immune responses determining the outcome of antigen exposure via the induction of CD4^+^ regulatory T cells (Tregs) and helper T cells (Th1, Th2, and Th17 cells, CD8^+^ cytotoxic T cells [13]) as well as unconventional γδ T cells [14]. Among these immune cell subsets, Th17 cells and γδ T cells can directly influence intestinal epithelial barrier functions by producing Th17-type cytokines and antimicrobial peptides [15]. Although both the small and large intestines comprise immune cells that can impact intestinal epithelial barrier functions, the distribution and abundance of these immune cells can differ between the two regions of the gut. For example, the small intestine is known to harbor a higher density of specialized immune cells called Peyer’s patches, which are essential for detecting and responding to luminal antigens, while the large intestine possesses a higher abundance of commensal bacteria that can interact with the host immune system [16,17]. Additionally, the types and amounts of cytokines and antimicrobial peptides produced by immune cells in the small and large intestines can differ depending on the local microenvironment and the specific microbial populations present. The difference between the bacterial density and composition of the small and large intestines can contribute to variations in the cytokines and antimicrobial peptides produced by immune cells in each region. Immune cells in the small intestine tend to produce higher levels of anti-inflammatory cytokines such as IL-10, and lower levels of pro-inflammatory cytokines such as IL-17, when compared to immune cells in the large intestine. In contrast, immune cells in the large intestine tend to produce higher levels of pro-inflammatory cytokines and antimicrobial peptides [18].

Reciprocally, intestinal bacteria have the ability to alter epithelial cytokine expression, thereby exerting control over the activity of T and B lymphocytes as well as macrophages [19,20]. Consequently, the balance between the ratios of epithelial barrier protective cytokines (IL-22, IL-17, and IL-33), pro-inflammatory cytokines (IL-1β, TNF-α, IL-2, IL-6, IL-15, IL-21, and IL-23) and anti-inflammatory cytokines (TGF-β, and IL-10) can determine the inflammatory or homeostatic state of the gut. Moreover, the gut microbiota produces a diverse range of metabolites, such as short-chain fatty acids (SCFAs), which have a protective role during inflammation. SCFAs like butyrate, acetate, and propionate can regulate the activity of different immune cells by blocking histone deacetylase (HDAC) in colonocytes and allowing histone hyperacetylation [21,22,23,24,25,26]. Notably, butyrate plays a crucial role in regulating immune function, epithelial barrier function, and intestinal homeostasis through multiple mechanisms such as facilitating tight junction assembly via the activation of AMPK, Akt, and other signaling pathways, acting as a signaling molecule for cell-surface G-protein-coupled receptors (GPRs) and nuclear factors (NFs) to interact with downstream effectors such as HDACs to reduce inflammation, and enhancing mucin secretion (reviewed by Recharia et al., 2023) [27]. Studies have proposed that butyrate is produced in higher amounts in the large intestine compared to the small intestine due to the higher bacterial density and greater availability of fermentable substrates in the colon [28]. Thus, the symbiotic interplay between gut microbiota and the gut mucosal immune system regulates the host’s immunological as well as metabolic function.

Viral infections, however, can disrupt the symbiotic interplay between gut microbiota and the mucosal immune system. Particularly, viruses targeting mucosal tissues such as Human Immunodeficiency Virus (HIV) [29], influenza [30], and severe acute respiratory syndrome coronavirus 2 (SARS-CoV-2) [31] are associated with varying levels of gut dysbiosis. Chronic HIV infection is linked to persistent inflammation, altered mucosal immune responses, impaired intestinal barrier integrity, and gut dysbiosis, which affects both the small and large intestine [15,32,33,34]. Influenza, an acute respiratory infection that disrupts pulmonary barrier integrity, is frequently correlated with extrapulmonary problems, including intestinal disorders that perturb the composition and function of the gut microbiota [35]. Some potential intestinal disorders that may be associated with influenza include inflammatory bowel disease (IBD), irritable bowel syndrome (IBS), and infectious diarrhea, which can affect both the small and large intestine. Similarly, coronavirus disease 2019 (COVID-19) caused by SARS-CoV-2 is associated with mild intestinal inflammation, alteration in intestinal barrier properties, and differences in the gut microbiota [36,37,38]. The altered gut microbiome and chronic inflammation have been shown to have far-reaching effects, not only on viral immunity but also on various vital organs, including the brain and heart. These disruptions can contribute to developing life-threatening disorders such as autoimmune diseases. Hence, the comprehensive understanding of the interaction between the gut microbiota and host immunity at the gut mucosal interface is critical for the development of therapeutic approaches toward ameliorating chronic inflammation during viral infections including, HIV, influenza, and SARS-CoV-2 infection [39,40].

This review aims to highlight the role of gut dysbiosis in susceptibility to virus acquisition and pathogenesis, with a specific focus on mucosal pathogens. It explores the interplay between gut microbiota and mucosal immune cells, particularly γδ T cells, in the context of viral pathogenesis and disease outcomes. The review also discusses the growing evidence and perspectives on therapeutic modulation of gut microbiome to alleviate the harmful effects of viral pathogenesis on host immunity.

## 2. Impact of Gut Microbiome on the Host Immune System under Steady State

It has been well established that while the gut microbiota stimulates and shapes innate and adaptive immune responses, the maturation of the gut microbial consortium is in turn affected by the host immune components [41,42]. A recent study by Wiertsema et al. has shown the crucial role of gut microbiota in the activity and development of intestinal immune cells [43]. The study demonstrated that germ-free (GF) mice display impaired development and maturation of gut-associated lymphoid tissues (GALT), smaller Peyer’s patches and mesenteric lymph nodes, and less cellular lamina propria of the small intestine, indicating an essential role for gut microbiota in the development and function of multiple populations of gut immune cells.

The organisms constituting the gut microbiota are typically separated from the host by the mucosal epithelium, which is composed of a single layer of tightly connected intestinal epithelial cells (IECs) and functions as a physical and chemical barrier [44]. IECs express receptors for the cytokines IL-17 and IL-22 produced by specialized tissue-resident lymphocytes including CD4 T, CD8 T, and innate lymphoid cells [45,46]. The dysregulation of IL-17/IL-22 cytokine secretion could lead to intestinal barrier disruption and the development of age-related inflammation [15], which is a likely factor underlying worse outcomes of viral infections in aging individuals. These tissue-resident lymphocyte subsets promote gut integrity through the induction of epithelial cell proliferation and inducing the expression of claudins, defensins, and mucin [47].

Furthermore, gut microbiota also contribute to intestinal homeostasis as their metabolites have been suggested to be involved in the modulation of immune responses in both adaptive and innate cells [48]. In addition, SCFAs can increase the expression of anti-inflammatory cytokines such as IL-10 and TGF-β and downregulate the level of pro-inflammatory cytokines in innate immune cells, leading to the inhibition of Th17 cell development and suppression of inflammation [21]. Moreover, through HDAC inhibition, SCFAs can modulate the homeostasis of peripheral T cells and Tregs development to control intestinal inflammation and carcinogenesis [26]. SCFAs can also activate G protein-coupled receptors (GPRs) including GPR41, GPR43, and GPR109A on colonocytes and immune cells and also induce anti-inflammatory pathways [23,24]. Based on studies, the mucin secretion and barrier integrity of gut mucosa can be provided by SCFAs through initiating the inflammasome and the peroxisome proliferator-activated receptor-γ (PPARγ) signaling pathway [21].

Figure 1 summarizes the mechanisms by which the gut microbiota influences host immunity and homeostasis, but it includes an additional factor, viral pathogens, to which we now turn our attention. Viral infections can also perturb host immunity and homeostasis with a consequent impact on the gut microbiota. This is particularly relevant for viruses that target intestinal and airway mucosa, such as HIV, SARS-CoV-2, and influenza A and B.

## 3. Role of Intestinal Immune Cells during Viral Dysbiosis

Intestinal immune cells are essential for maintaining immune homeostasis and defending against viral infections. In the small intestine, the gut microbiota is generally less abundant than in the large intestine, and the immune response is more focused on preventing the invasion of pathogens. In contrast, the colon has a higher abundance of gut microbiota, which plays a key role in maintaining gut homeostasis and promoting immune tolerance as reviewed in detail by Lee et al. (2022) [49]. In the case of viral infection, the immune response in the large intestine is more focused on controlling the expansion of the virus and reducing inflammation. This is achieved by the activation of regulatory T cells, which aid in suppressing overactive immune responses, and by the generation of antimicrobial peptides by the gut’s epithelial cells, which contribute to controlling viral replication. During viral dysbiosis, immune cells APCs such as DCs and macrophages as well as immune effectors such as T cell subsets and innate lymphoid cells (ILCs) are involved in several essential functions [50]. Virus-specific antibody production by B cells in the GALT, including IgA, can neutralize viruses and prevent their attachment to intestinal epithelial cells [51]. The more recently discovered ILCs, which rely on signals from the gut microbiome for their phenotypic diversity and functional plasticity [52], secrete polarized cytokines and chemokines to fight infection and promote epithelial barrier repair [53]. Over the last decade, extensive research has improved the understanding of mechanisms regulating ILC plasticity in response to mucosal pathogens [54].

Among various intestinal immune cells, γδ T cells are a major effector cell subset found in the gut lamina propria and IEL compartments, and they are involved in preserving the balance of the gut environment by releasing specific cytokines, such as IL-22 and IL-17, and growth factors, including keratinocyte growth factor (KGF). These factors contribute to the integrity of the gut epithelial barrier and also stimulate the production of antimicrobial peptides (AMPs) by gut epithelial cells in the small intestine, thus suppressing microbial populations [55]. Moreover, γδ T cells play a protective role in immune surveillance by recognizing a unique and wide array of danger signals and rapidly detecting viral infection [56]. γδ T cells can directly kill infected cells by diverse pathways, including the secretion of cytotoxic mediators collected in granules such as perforin and granzymes (A, B, and M); via the expression of members of the death-inducing TNF family of ligands and receptors, including tumor necrosis factor-related apoptosis-inducing ligand (TRAIL); the natural killer group 2, members C and D (NKG2C, NKG2D)-mediated cytotoxicity; antibody-dependent cell-mediated cytotoxicity (ADCC), and the apoptosis pathways triggered by death inducible receptors like FAS [57,58]. γδ T cell-mediated non-cytolytic antiviral activity is mediated via cytokines and chemokine production following activation by virus-infected cells [59].

Notably, the homeostasis of γδ T cells has been linked with the stability of host commensal microbiota. Ligand binding to TLRs on γδ T cells through pathogen-associated molecular patterns–toll-like receptors (PAMP-TLR) triggers an activating effect within the myeloid differentiation factor 88 (MyD88) pathway [34]. The integrated signals of TLR3 and TCR play an antiviral effect in γδ T cells supporting the critical role of γδ T cells in viral infections. At the same time, TLR8 activation can reverse the anti-inflammatory action of γδ T cells [32,56]. Furthermore, intestinal γδ T cells were shown to modulate gut microbiota and fecal micro-RNAs to maintain mucosal tolerance in mice [60]. Therefore, γδ T cells are a double-edged sword in intestinal inflammation and protection, and the mechanism of the interaction between the gut microbiota and γδ T cells in the lamina propria should be clarified [61].

Overall, intestinal immune cells play a critical role in detecting, eliminating, and controlling viral spread during viral dysbiosis via collaboration with epithelial cells and host microbiota to maintain the integrity of the intestinal barrier. This barrier prevents the translocation of viruses from the gut lumen into the bloodstream and other tissues. Their coordinated responses are essential for maintaining gut health and preventing the systemic spread of viral pathogens.

## 4. Gut Dysbiosis during Viral Infections

### 4.1. HIV/SIV

HIV infection, primarily targeting the gastrointestinal tract for transmission and early replication, leads to significant changes in the gut mucosa and contributes to pathogenesis [62]. Blood and gut mucosal CD4^+^ T-cell depletion is a hallmark of HIV infection that affects both the small and large intestine [63], owing to viral infection via the CD4 receptor, the depletion of Th17 cells [64], chronic immune activation [65], and an alteration in cell types responsible for maintaining IL-17-producing cells, such as gut-resident APCs [66]. However, the degree of depletion may be greater in the large intestine, which is believed to be due to higher levels of viral replication and immune activation in this region [67]. Furthermore, the depletion of CD4+ T cells in the gut mucosa during HIV infection can lead to a reduction in the number of immune cells that regulate the composition and function of the gut microbiota, potentially contributing to dysbiosis and further altering the gut environment. GI tract dysfunction following HIV/Simian immunodeficiency virus (SIV) infection can cause a progressive impairment of mucosal barrier integrity, which is correlated to functional alterations in local leukocytes such as those expressing IL-17 and/or IL-22 [68]. The early initiation of antiretroviral therapy (ART) is associated with the partial restoration of IL-17-producing cells in the GI tract during HIV-1 infection [69]. Although the mechanisms by which the loss of these IL-17/IL-22-producing cells occurs are still unknown, inflammatory mediators could play a crucial role.

HIV infection is correlated with reduced gut microbiota diversity and richness in both the small and large intestines, but the degree of reduction may be greater in the large intestine. Additionally, studies have suggested that the composition of the gut microbiota may be more profoundly altered in the large intestine, with a greater reduction in beneficial bacterial species such as *Lactobacillus* and *Bifidobacterium*. This may be due to the fact that the large intestine provides a more favorable environment for bacterial growth and colonization than the small intestine [70].

Numerous studies confirming significant gut dysbiosis during HIV infection [71,72,73,74] have been performed in human cohorts comparing HIV-infected gut microbiota composition to that of HIV-uninfected control subjects [40,68]. These studies have revealed several common patterns among HIV-infected persons, including an increased relative abundance of *Erysipelotrichaceae*, *Enterobacteriaceae*, *Desulfovibrionaceae* and Fusobacteria and a decreased abundance of *Lachnospiraceae*, *Ruminococceae*, *Bacteroides* and *Rikenellaceae.* Recently, it was reported that the gastric microbiota of HIV-1-infected individuals when compared to HIV-uninfected individuals had an increased relative abundance of *Proteobacteria* and decreased abundance of *Bacteroides,* which are bacteria associated with alleviating inflammation [75]. Although contradictory findings have been reported regarding the *Prevotella* genus, some studies observed that *Prevotella* relative abundance was increased in chronic HIV-1 infection [76,77]. In a study of HIV-infected persons, Dubourg et al. revealed that Ruminococcaceae and Faecalbacterium, especially *Ruminococcus bromii* and *Faecalibacterium praustintzii*, were lower than healthy subjects and inversely correlated with inflammation and immune activation markers [78]. Our studies in the nonhuman primate model of HIV infection have also demonstrated longitudinal changes in gut microbial species consistent with reports in the chronic HIV infection [40]. Specifically, an increased abundance of *E. rectale* during chronic infection with antiretroviral treatment were significantly correlated with inflammatory cytokines, and an abundance of *F. praustintzii and Treponema succinifaciens* was negatively correlated with inflammatory cytokines and leaky gut biomarkers [40].

The gut microbiome composition among men who have sex with men (MSM) and HIV-infected populations is characterized by a *Prevotella*-rich microbiome and a decreased abundance of *Bacteroides* [77]. HIV-infected MSM subjects were found to have elevated levels of fecal calprotectin (which is a biomarker of GI inflammation) as well as inflammatory fecal soluble immune factors (sIFs) (GM-CSF, ICAM-1, IL-1β, IL-12/23, IL-15, IL-16, TNF-β, VCAM-1, and VEGF) and decreased levels of beneficial cytokines IL-22 and IL-13 [79]. Based on both clinical and in vivo experiments, there is a significant change in enteric viral diversity during HIV-1 and progressive SIV infection in addition to bacterial dysbiosis [80,81]. The alteration in viral diversity of SIV-infected Asian macaques, including the increased abundance of pathogenic adenoviruses, is associated with lesions in the intestinal epithelium and low peripheral CD4^+^ T-cell counts [82]. Furthermore, some studies have linked dysbiosis with HIV/SIV pathogenesis and essential features in disease progression, such as the activation of CD4^+^ and CD8^+^ T-cells in the blood and gut mucosa of HIV-1-infected individuals [83] and an increase in inflammatory and leaky gut biomarkers [40].

Taken together, these studies outline the increasing importance of the microbiome composition in HIV-1 pathogenesis and leave open the possibility of therapeutic strategies aimed at targeting the microbiome to improve the health of HIV-1-infected subjects.

### 4.2. Influenza

In addition to causing respiratory tract disease, viral infections such as influenza can also cause extrapulmonary manifestations, including intestinal disorders [84]. In vivo studies in murine models have shown that severe infections with H1N1 and H5N1 influenza A virus (IAV) were accompanied by an alteration of the gut microbiota as well as enhanced susceptibility to secondary enteric infections [30]. Therefore, the nature and consequences of influenza-associated dysbiosis on clinical outcome remain an active area of research. Recent studies have shown that influenza alters the composition and metabolic activity of the gut microbiota, leading to immunological dysregulation that could increase inflammatory gut disorders [85,86]. Preclinical and clinical studies have indicated that influenza infections in murine models (H1N1 and H5N1 IAV) and in humans (H7N9) alter the composition of the gut microbiota [85,87,88,89]. Sencio et al. revealed that infection with the H3N2 and H1N1 IAV is associated with changes in the gut microbiota composition, resulting in a decreased production of fermentative, gut microbiota-derived products SCFA [30] such as acetate and favor the secondary pneumococcal infection of the lungs. Furthermore, the reduced production of SCFA by influenza virus infection could impair the gut barrier integrity and support secondary enteric infections [35]. SCFAs are also considered to maintain intestinal barrier integrity by stimulating the synthesis of antimicrobial peptides against invading enteric pathogens [90].

Evidence suggests that gut dysbiosis can exacerbate the severity of influenza virus infections [91], supporting the concept of gut dysbiosis contributing to impaired immune function and increased susceptibility to viral infections [92]. Studies have suggested that influenza infection can lead to alterations in the immune response in the gut, specifically affecting the population of immune cells called intraepithelial lymphocytes (IELs). Influenza infection can cause a reduction in the number of IELs in the small intestine but not in the large intestine. This reduction in IELs may contribute to impaired immune function in the gut [84]. In particular, dysbiosis-induced gut alteration could lead to an impaired intestinal barrier, allowing bacterial products such as lipopolysaccharides to escape into the bloodstream and initiate a systemic inflammatory response [93], which could worsen the severity of influenza infection by exacerbating lung inflammation and increasing virus replication [94]. For instance, H9N2 avian influenza virus infection has been reported to disrupt microbial homeostasis and induce inflammatory damage in the intestinal mucosa and gut dysbiosis [95,96]. Additionally, H9N2 AIV infection could trigger an increase in the expression of pro-inflammatory cytokines and increase the abundance of proteobacteria such as *Escherichia coli* while decreasing the abundance of *Lactobacillus* and *Enterococcus* [95,96].

### 4.3. SARS-CoV2

SARS-CoV-2 has emerged as another critical respiratory virus that is associated with gut dysbiosis. The major entry receptor, angiotensin-converting enzyme 2 (ACE2), a type I integral metallocarboxypeptidase, is found mainly expressed at high levels in epithelial and endothelial cells in renal, kidney, lung parenchyma, and the GI tract [97]. The high expression of ACE2 on the small bowel enterocyte brush border and co-expression of TMPRSS2 supports potential enteric infection by SARS-CoV-2 [98,99]. Following the viral entry into the host cell, single-stranded RNA is recognized by the innate immune system via PRRs such as TLR7 and TLR8, RIG-I (retinoic acid-inducible gene I)-like receptors (RLRs), and the NOD-like receptor, CARD-containing-2 (NLRC2) [100]. In addition to the secretion of type-I and -III antiviral interferons (IFNs) and chemokines during early response [55], pro-inflammatory cytokines and chemokines are also produced by epithelial cells during SARS-CoV-2 infection, leading to the characteristic “cytokine storm” and ARDS of severe COVID-19 [101].

An emerging line of evidence suggests that gut dysbiosis during SARS-CoV-2 infection contributes to the severity of the disease by disrupting the immune response and increasing the production of pro-inflammatory cytokines [102,103,104]. The balanced gut microbiome has an immunomodulatory role on the host by maintaining a healthy equilibrium between pro-inflammatory and anti-inflammatory responses. Gut dysbiosis could disrupt this balance and produce an exaggerated immune response, resulting in tissue damage and organ dysfunction during severe COVID-19. One study found that SARS-CoV-2 infection could lead to a decline in the abundance of beneficial microorganisms such as *Faecalibacterium prausnitzii* and *Bifidobacterium* in the large intestine but not in the small intestine. The study additionally reported that the level of gut dysbiosis was linked with disease severity, with more severe cases of COVID-19 showing greater alterations in the gut microbiota [103]. Accordingly, dysbiosis has been related to the severity of disease, as individuals experiencing more severe symptoms show more significant alterations in the gut microbiome composition [103,104]. This was further confirmed in a study using transgenic mice expressing human ACE2 that reported SARS-CoV-2-mediated perturbation of the gut microbiota structure in a dose-dependent manner upon intranasal inoculation [36,103,104].

Although the exact mechanisms underlying the relationship between gut dysbiosis and COVID-19 are still not fully understood, studies have confirmed infectious viral particles and SARS-CoV-2 replication in the rectal mucosa [105,106]. Further investigation is needed to unravel the complex interplay between the gut microbiome, SARS-CoV-2, and the immune system to clarify the role of gut microbiota in COVID-19 pathogenesis and long COVID.

## 5. Role of Host Microbiota in Susceptibility to Viral Infection and Pathogenesis

The intricate crosstalk between the commensal microbiota and the immune system can contribute to developing or suppressing infection [107]. Evidence suggests that host microbiota can influence immunity at mucosal surfaces against a range of viral infections via AMP secretion [108,109], the inhibition of viral attachment to host cells [110,111,112], and the modulation of mucosal immune cells [113,114]. In a study by Kim et al., *Staphylococcus epidermidis*, a common human nasal commensal, was shown to suppress the replication of the influenza A virus in the nasal mucosa of mice [115]. This suppression prevented IAV spread to the lung and was caused by the stimulation of interferon (IFN) innate immunity. Similarly, Ji et al. have demonstrated suppressed RSV infection and reduced lung pathology in neonatal mice by modulating gut bacteria by oral treatment with specific strains of bacteria including *Escherichia coli, Streptococcus thermophilus*, *Bifidobacterium* spp. and *Lactobacillus* spp. [116].

On the other hand, several studies have tested the impact of the microbiome on pathogenic viruses by studying the infection of hosts after microbiome depletion with antibiotic treatment. Microbiome depletion studies in mice [117] and chickens [118] have shown that the microbiota drives an IFN response in the lungs that stops early IAV replication. Depletion of the vaginal microbiome in mice was shown to increase IL-33 production, which suppresses IFNγ secretion, leading to increased Herpes Simplex Virus type 2 (HSV-2) susceptibility [119]. Additionally, vaginal microbial communities dominated by *Lactobacillus crispatus* were associated with a decreased HIV infection in South African women [120]. Several *Lactobacillus* species (*L. crispatus, L. gasseri, and L. vaginalis*) have been shown to inhibit HIV-1 replication in ex vivo cervico-vaginal tissue culture. These effects are mediated through acidification of the medium and lactic acid production, as well as their binding to the virus, thereby reducing free virions in the tissue [121].

Even though the microbiota plays a protective role in preventing or fighting invading pathogens, it may also enhance viral pathogenesis. Differences in the distal intestinal microbiota are reported in respiratory and hepatitis virus infections, supporting the theory that the gut microbiota acts systemically rather than locally in viral infections [122]. Based on a study by Fulcher et al., pre-existing microbiome alterations (such as decreased abundance of *Bacteroides*) could contribute to a chronic inflammatory state which heightens HIV susceptibility and acquisition [123]. The authors suggested that microbiota-associated systemic inflammation, characterized by elevated cytokines/biomarkers and bioactive lipids, could be involved in increased susceptibility to HIV infection. The study’s findings highlighted the importance of the gut microbiota and mucosal immune system in HIV pathogenesis and proposed that interventions aimed at restoring healthy gut microbiota and reducing systemic inflammation could be potential strategies to prevent or treat HIV infection.

These studies demonstrate that commensal bacteria in different mucosal sites can influence susceptibility to pathogenic viruses. Nevertheless, more research is essential to define their role in antiviral and pathogenic mechanisms.

### 5.1. Microbiome–Gut–Brain Axis in Viral Neuropathogenesis

Intestinal homeostasis is intricately connected to the central nervous system (CNS) through the physiological contributions of gut microbiota, regulation of intestinal barrier function, and activity of peripheral neurons. Referred to as the “microbiome–gut–brain axis”, the communication between the gut and brain has recently garnered much interest in the context of long-term neurological symptoms of COVID-19 [124], which include chronic fatigue, anxiety, olfactory and gustatory dysfunction, dementia, depression, anxiety, memory impairment, stroke, and encephalitis.

The connection of viral dysbiosis with neuropathogenesis has been previously established in the setting of chronic HIV infection, which is associated with increased burden of neurocognitive disorder, dementia, depression, and substance use [125]. The complex bidirectional communication pathway between the GI tract and CNS involves a network of neural, hormonal, and immunological pathways that allow for coordinated responses to changes in the gut environment, such as exposure to pathogenic viruses. For instance, gut dysfunction and systemic inflammation driven by chronic HIV infection lead to neurological symptoms such as cognitive impairment, depression, and anxiety [124,126]. Furthermore, the development of HIV-associated neuroinflammation and cognitive impairment has been directly linked to gut dysbiosis and increased intestinal permeability [127]. It has been proposed that HIV-induced alterations in the balance of neurotransmitters in the brain contributes to disruption of the gut–brain axis, which can lead to the development of depression and other neuropsychiatric symptoms [128].

In the context of SARS-CoV-2 infection, recent studies have found that COVID-19 patients with neurological symptoms have altered gut microbiota compared to those without neurological symptoms [129]. In addition, SARS-CoV-2 has been found in the gut of some infected patients, suggesting that the gut may serve as a reservoir for the virus [103]. These findings support the concept of the gut–brain axis playing a role in the development of COVID-19-associated neurological symptoms. Several mechanisms have been proposed to explain the role of the gut–brain axis in the development of viral neuropathogenesis. Alterations in the gut microbiome and gut barrier function can translocate microbial products, such as LPS, into the blood circulation, which can activate immune cells and induce neuroinflammation. Inter-related mechanisms may contribute to loss of gut barrier integrity and breach of the blood–brain barrier, thereby linking gut dysbiosis to neuroinflammation.

Further research is necessary to fully reveal the exact mechanisms underlying the gut–brain axis in the development of neuropathogenesis during viral infections, including HIV, CMV, and SARS-CoV-2. Potential interventions that target the gut microbiome and gut barrier function may hold promise for preventing or treating neurological complications linked to these infections. For example, strategies aimed at restoring a healthy gut microbiome and improving gut barrier function could help reduce the translocation of microbial products and the activation of immune cells in the gut mucosa, thereby mitigating the development of neuroinflammation and neuropathogenesis in these infections. It is worth mentioning that the gut–brain axis is a manifold and multifactorial system, and its interactions with viral infections may be influenced by various factors, including host genetics, age, sex, and comorbidities. Therefore, interventions that target the gut–brain axis in the context of viral infections may need to be individualized and tailored to the specific needs of each patient.

### 5.2. Cardiovascular Disease Associated with Dysbiosis during Chronic Viral Infections

Chronic inflammation and viral infection have been independently associated with the development of cardiovascular disease (CVD) [130]. Dysbiosis can promote inflammation and oxidative stress, which are key cardiovascular disease drivers [131]. Increasing evidence points to the role of gut dysbiosis during chronic viral infections in the development of CVD [132]. This highlights the importance of understanding the role of viruses in the pathogenesis of atherosclerosis toward developing strategies to reduce the risk of CVD.

Chronic HIV infection, for instance, is correlated to an increased risk of developing cardiovascular diseases, such as myocardial infarction, cerebrovascular disease, and peripheral vascular disease, due to enhanced atherosclerosis and peripheral vascular disease [133]. It has been demonstrated that the gut microbiota of HIV-negative individuals who have increased the metabolism of choline into trimethylamine (TMA), which is then oxidized to trimethylamine-N-oxide (TMAO), may have atherogenic potential [134]. The gut dysbiosis observed in HIV-positive individuals has been attributed to an increase in *Prevotella*, a bacterial genus known to produce TMA, which has been found to be elevated in HIV-positive individuals [135]. Alternatively, gut dysbiosis may result from a depletion of beneficial bacterial species, such as *Akkermansia*, which has been recognized to possess a protective role against metabolic disorders [70]. Moreover, impaired gut integrity can promote lipopolysaccharides (LPS) translocation into the bloodstream and further exacerbate systemic inflammation in cardio-metabolic diseases and endothelial dysfunction [136]. Endothelial dysfunction is a key variable in the pathogenesis of atherosclerosis, which is a chronic inflammatory condition that underlies many cardiovascular diseases [137].

In general, gut dysbiosis during chronic viral infections can contribute to the development of cardiovascular disease probably through several mechanisms, including inflammation, oxidative stress, endothelial dysfunction, and impaired SCFA production. Future research may investigate the potential of interventions such as probiotics, prebiotics, or fecal microbiota transplantation (FMT) to modulate the gut microbiome and improve outcomes in patients with chronic viral infections and cardiovascular disease.

### 5.3. Fatty Liver Disease Due to Viral Dysbiosis

Fatty liver disease (FLD) is a condition characterized by the accumulation of fat in the liver, which can lead to inflammation and damage to liver cells. Emerging evidence suggests that gut dysbiosis during viral infections can promote the development of fatty live FLD [138]. Viral infections such as HBV and HCV have been associated with changes in gut microbiota composition and function, leading to dysbiosis [139]. Dysbiosis can promote the translocation of bacterial products such as LPS, which are also known as endotoxins into the liver, leading to inflammation and the development of FLD. Studies have shown that the gut microbiome plays a crucial role in regulating hepatic lipid metabolism [140]. Dysbiosis during viral infections can lead to impaired lipid metabolism, promoting fat accumulation in the liver [141]. In addition, dysbiosis can also promote the production of pro-inflammatory cytokines, such as tumor TNF-α, which can further exacerbate liver inflammation and damage [142].

Notably, dysbiosis is one of the factors underlying the pathogenesis of nonalcoholic fatty liver disease (NAFLD) in HIV-infected persons, contributing to a higher risk of CVD [143]. A recent follow-up study in COVID patients described a high prevalence of metabolic (dysfunction)-associated fatty liver disease (MAFLD) with a potential to accelerate CVD in long-COVID-19 [144]. Although FLD prevalence in long-COVID-19 patients in this study was more than double that in the general population, the study lacked data on the presence of MAFLD before COVID-19 and thus cannot rule out the possibility of metabolically unhealthy individuals being over-represented in the cohort. It is possible that the interaction between gastrointestinal microbiota and the lungs, the so-called “gut–lung axis”, is perturbed by SARS-CoV-2 infection and may contribute to metabolic dysfunction in long-COVID. More in-depth investigations into the mechanisms underlying metabolic dysfunction in persistent viral infections are needed to develop novel diagnostic and therapeutic measures to prevent or slow the acceleration of metabolic disease, such as targeting gut dysfunction and dysbiosis.

## 6. Microbiome-Based Therapies in Viral Infections

Since gut dysbiosis could impair the host’s immune responses, resulting in worsening the pathogenicity of the virus, interventions aimed at restoring the balance of gut microbiota, such as probiotics or FMT, have the potential to alleviate the severity of viral infections like influenza [96]. Probiotic supplementation in HIV has been shown to improve the count of CD4+ T cells and alleviate HIV-induced illnesses, such as diarrhea and nausea [145]. Clinical studies have investigated the beneficial effects of probiotic supplementation on adult volunteers against common cold and influenza-like respiratory infections [146] and human influenza A/H1N1 and A/H3N2 viruses [147], and they have showed significantly higher levels of serum IFN-γ and secretory IgA in the gut and saliva. Probiotic supplementation has also proven beneficial in coronavirus infections; for example, transmissible gastroenteritis coronavirus (TGEV) in pigs is inhibited by *Enterococcus faecium*, *Lactobacillus plantarum*, and *Lactobacillus salivarius* supplementation [148,149].

Microbiota modulation via FMT is well established as an effective therapeutic option in the treatment of *C. difficile* infection (CDI) [150]. It is being explored for infectious viral diseases, including HIV and influenza virus [151]. In one trial involving 30 participants with HIV, FMT was found to reduce a biomarker of gut permeability [151]. Another smaller study, which involved weekly FMT in six people living with HIV, demonstrated increased gut microbial diversity over a 6-week treatment period. One participant in this study also experienced an improvement in biomarkers of gut permeability and inflammation [152]. Moreover, FMT in chickens improved resistance against the avian influenza virus [118]. The study demonstrated that modifying the composition of the microbiome through the use of probiotics and/or FMT could promote the development of a healthy microbiota that confers protection against influenza virus infection in chickens. In another study by Gao et al., FMT was utilized to regulate influenza-infected mice previously treated with antibiotics that had developed intestinal dysbiosis [153]. The study’s outcomes revealed that influenza-infected mice with antibiotic-induced intestinal dysbiosis experienced more severe damage to their lung tissue and intestinal mucosa than mice infected with the virus alone. In addition, the researchers also found that improving the intestinal flora through the use of probiotics or FMT could alleviate intestinal inflammatory responses and improve pulmonary inflammation via the TLR7 signaling pathway.

Overall, FMT has been demonstrated to restore the diversity of gut microbiota, improve intestinal barrier function, alleviate inflammation, and facilitate the restoration of immune functions. However, more research is needed to determine the safety and efficacy of FMT to mitigate concerns of adverse effects in immunosuppressed individuals due to the possible transmission of pathogens [154]. In this regard, Biliński et al. have reported that FMT appears to be safe and effective in treating recurrent CDI in patients with coexisting COVID-19 [155]. Despite the effectiveness of FMT, further preclinical and clinical studies should be conducted to clearly assess the safety of FMT. During the coronavirus pandemic, stool banks conducted intensive screening protocols for FMT donors, including serological and stool tests for several bacteria, viruses, and parasites, including Hepatitis (A, B, C, E) HIV, human T-lymphotropic virus (HTLV), and *Treponema pallidum* [156,157]. These interventions could also be potential therapeutic strategies for FLD due to viral dysbiosis [158]. They have been shown to improve gut microbiome composition and function, reduce inflammation, and improve liver function in animal models of FLD [159]. Nevertheless, more research is necessary to determine the optimal timing, dosage, and duration of such an intervention.

## 7. Conclusions and Future Directions

The global burden of viral infections remains alarmingly high in terms of mortality and morbidity. Viruses primarily enter the body through mucosal sites inhabited by commensal microbiota. Within these sites, viruses interact with the host microbiota, resulting in alterations to the host’s immune responses. The gut microbiota actively intervenes in viral infections to maintain the stability and equilibrium of mucosal sites, thus ensuring its own survival. On the other hand, pathogenic viruses have evolved mechanisms to exploit host microbiota to favor viral spread and immune evasion. Several studies indicate that the gut microbiome serves a dual role as both guardian and facilitator in the context of pathogenic viral infections within the gut. Therefore, it is vital to consider the complex interaction between viruses, microbiota, and immune cells within the host tissue microenvironment to understand how they determine disease outcomes.

In this review, we discussed studies that have demonstrated that probiotics and their metabolites possess the ability to decrease the risk of viral infections. Additionally, beneficial bacteria in the gut can act as adversaries against viral infections that disrupt gut homeostasis. We highlighted the role of mucosal γδ T cells as a significant immune cell subpopulation involved in both immune responses to viral infections and the regulation of mucosal tolerance and host microbiota composition. Overall, the gut microbiota is linked to viral infections and pathogenesis, and its modulation has excellent therapeutic potential. Future research endeavors should prioritize establishing the causal relationships between viral infections and gut dysbiosis. Moreover, it is crucial to develop standardized gut microbiota patterns among healthy individuals, considering the influence of key factors such as diet, lifestyle, and genetics on the gut microbiome. Also, determining the optimal routes for microbiome modulatory therapies, such as oral ingestion versus infusion via colonoscopy, in relevant preclinical models (including nonhuman primates) is of utmost importance. Likewise, standardizing screening measures to reduce pathobionts, including latent viruses, Enterobacteriaceae, and enterococci, from fecal samples is imperative for the development of microbiota transplantation as a therapeutic approach for viral disease pathogenesis.

In conclusion, the interactions between the gut microbiome and mucosal immune cells, including γδ T cells, have important implications for susceptibility and response to viral infections. Advancing our understanding of the complex mechanisms underlying these interactions is crucial, as it can pave the way for exploring the potential of microbiome modulation therapy as an effective strategy against viral pathogenesis. Further research is needed to delve deeper into these areas and uncover novel approaches to combat viral pathogenesis.

## Figures and Tables

**Figure 1 microorganisms-11-01849-f001:**
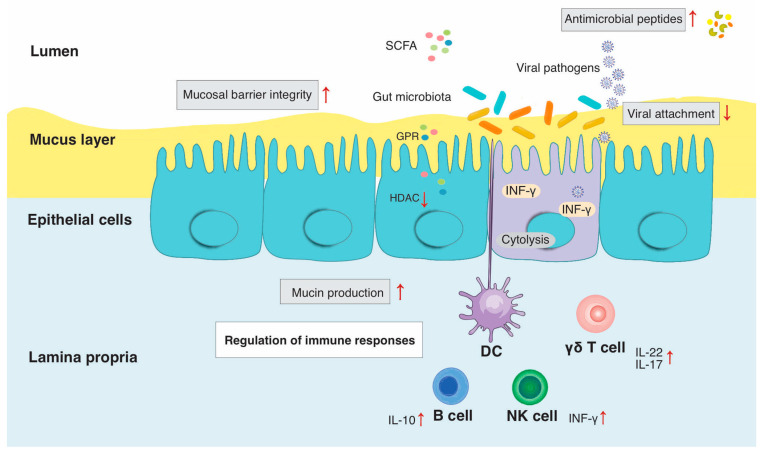
Interaction between host immunity, gut microbiota and viral pathogens in gut mucosa. Abbreviations: intraepithelial lymphocytes (IELs), innate lymphoid cells (ILCs), short-chain fatty acids (SCFAs). The gastrointestinal epithelium is composed of a single-cell layer that separates the intestinal lumen from the underlying lamina propria. Host immune cells including dendritic cells, γδ T cells, αβ T cells, ILCs, macrophages and B cells survey the IELs are interspersed throughout the epithelium and are positioned both within and directly below the epithelium. The mucus layer, containing mucins, provides a physical barrier between pathogens and host epithelia. Gut microbiota can influence mucin production with potential antiviral properties. Gut microbes including lactic acid bacteria can regulate tight junctions and thus maintain normal mucosal permeability and produce antimicrobial compounds (bacteriocins) that can inhibit viral attachment. Lastly, gut microbiota can modulate immune cell functions through the production of SCFAs in the small intestine and stimulate γδ T cell and ILC functions.

## Data Availability

Not applicable.

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
