# Peer review of "The Role of Gut Dysbiosis in the Loss of Intestinal Immune Cell Functions and Viral Pathogenesis"

_microorganisms, 2023, doi:10.3390/microorganisms11071849_

Round 1

Reviewer 1 Report

The article discusses important and current topics related to intestinal microbiota dysbiosis and viral infections. The authors focused mainly on three viruses (HIV, influenza and SARS-CoV-2), although they also cite the results of research on others. I have only one concern about this review article – the Authors do not distinguish between the small and large intestine. These two parts of the gastrointestinal tract differ in terms of the functions, composition of the inhabiting microbiota and the profile of bacterial metabolites that impact the intestine functioning. I recommend specifying to which part of the intestine the Authors refer particular effects/ observations – it is of special importance (but is not limited solely to) when aspects specific for only one of the intestine parts (e.g. antibacterial peptides production in the small intestine) is discussed.

Minor comments

Line 104. please delete “influencing”

Line 147 please explain “ILCs” at the first use

Line 189 “and” please normal font (not italics)

Line 195 please explain “sIFs” at the first use

Line 153 and 287-289 please be precise and indicate that antimicrobial peptide production by epithelial cells takes place in the small intestine

Line 321 please indicate that references 101 and 101 concern herpes virus (thus not those you focused on in the review)

Line 328 do you mean “a mixture of …”?

Line 430 please explain “FMT” at the first use

Line 437-439 please specify which bacterial metabolites contribute to this effect

Please revise the references list, as the position “BiliÅ„ski et al.” (line 494) is missing.

Author Response

We thank the reviewer for pointing out the importance of differences in the microbiota and immune functions in the small and large intestine. We have now included this in various sections throughout the manuscript along with relevant references.

We have addressed all the minor comments with highlighted text and revised the references list. 

Reviewer 2 Report

Significance and novelty and attractiveness to readers

This searches exactly the intrinsic relationships and interaction between some factors and players, affecting the main axe, microbiome - overall health. The article reciprocal interactions between dysbiosis, which can profoundly affect various biological aspects, including susceptibility to viral infections and simultaneously is investigating the influence of viral infections on the gut microbiome. So, this reciprocal influence of gut dysbiosis and viral infection and pathogenesis is quite an interesting area of significant needed research. Moreover, the study casts light on the influence of gut microbiome on the host immune system by regulating intestinal homeostasis through interactions with immune cells.

Thus, this review may enhance our understanding of how viral infections perturb the gut microbiome and mucosal immune cells, affecting in turn host susceptibility and submission to viral infections. The study focused specifically on exploring the interactions between gamma delta (γδ) T cells and gut microbes within the context of inflammatory viral pathogenesis.  Additionally, the study discussed emerging evidence and potential future directions for microbiome modulation therapy in the context of viral pathogenesis which is a very important topic for future experimental studies.

So, the subject of this article is quite novel, significant, inspiring, and inseminating.

Article structure

The structure of the article is as follows:

1. Introduction

2. Impact of gut microbiome on influencing the host immune system under steady state 1

3. Gut dysbiosis during viral infections

3.1. HIV/SIV

3.2. Influenza

3.3. SARS-CoV2

4. Role of intestinal immune cells during viral dysbiosis

5. Role of host microbiota in susceptibility to viral infection and pathogenesis

5.1. Microbiome-gut-brain axis in viral neuropathogenesis

5.2. Cardiovascular disease associated with dysbiosis during chronic viral infections.

5.3. Fatty liver disease due to viral dysbiosis

6. Microbiome-based therapies in viral infections

7. Conclusions and Future Directions

Sequentially and logically the entry 4 should come after the entry 3.

So the sequential structure may be better presented as follows:

1. Introduction

2. Impact of gut microbiome on influencing the host immune system under steady state 1

3. Role of intestinal immune cells during viral dysbiosis

4. Gut dysbiosis during viral infections

4.1. HIV/SIV

4.2. Influenza

4.3. SARS-CoV2

5. Role of host microbiota in susceptibility to viral infection and pathogenesis

5.1. Microbiome-gut-brain axis in viral neuropathogenesis

5.2. Cardiovascular disease associated with dysbiosis during chronic viral infections.

5.3. Fatty liver disease due to viral dysbiosis

6. Microbiome-based therapies in viral infections

7. Conclusions and Future Directions

Figures

Figure 1 title should be concise and self-explained (Figure 1. Interaction between host immunity, gut microbiota and viral pathogens in gut mucosa.). So, all the abbreviation appearing on the figure should be explained after the title directly.

However, the text from L145 to L155, should be separated from the title and be presented as an independent paragraph.

Minor linguistic modifications

L36, change (extensive and diverse) into (an extensive and diverse)

L36-37, correct (microoganisms) into (microorganisms)

L38, change (Importantly) into (Notably).

L39, change (a range of) into (various).

L40, change (various aspects of human health) into (aspects of human health)

L49, change (serves as) into (is).

L51, change (specifically targets pathogens) into (targets pathogens specifically)

L56, change (that determine the outcome) into (, determining the outcome).

L61, change (In a reciprocal manner,) into (Reciprocally,)

L67, change (such as) into (, such as)

L67-68, change (short- chain-fatty acids) into (short- chain fatty acids)

L68, change (that have) into (, that have)

L81, change (, severe acute) into (, and severe acute)

L85, change (integrity) into (integrity, )

L92, change (can contribute to the development of life-threatening) into (can contribute to developing life-threatening)

L96, change (including HIV) into (, including HIV)

L106, change (in turn) into (, in turn)

L107, change (et al,) into (et al.)

L108, change (on the activity) into (in the activity)

L115, change (, which is composed of a single) into (, composed of a single)

L118, change (including) into (, including)

L122, change (through induction) into (through the induction)

L132, change (including) into (, including)

L141, change (viruses the target) into (viruses that target)

L158, change (, which primarily targets) into (, primarily targeting)

L164, change (integrity which) into (integrity, which)

L181, change (HIV infected) into (HIV-infected)

L185, change (species that are consistent with reports in chronic) into (species consistent with reports of chronic)

L202, change (important features) into (essential features)

L202, change (progression such as) into (progression, such as)

L204, change (and increase in inflammatory) into (and an increase in inflammatory)

L209, change (respiratory viral infections) into (viral infections)

L201, change (including) into (, including)

L214, change (remains) into (remain)

L240, change (important) into (critical)

L257-258, change (could lead to disruption of this balance) into (could disrupt this balance)

L260, change (related with the severity of disease,) into (related to the severity of the disease,)

L261, change (show greater alterations) into (show more significant alterations)

L265, change (Although, the exact) into (Although the exact)

L268, change (between gut microbiome) into (between the gut microbiome).

L272-275, please revise carefully this sentence.

L275, change (several important functions) into (several essential functions)

L280, change (Extensive research over the last decade has improved the) into (Over the last decade, extensive research has improved the)

L283, change (a prominent) into (a major)

L291, change (can directly kill of infected cells) into (can directly kill infected cells)

L305, change (infections, while TLR8 activation) into (infections. At the same time, , TLR8 activation)

L311-312, change (in the detection, elimination, and control of viral spread) into (in detecting, eliminating, and controlling viral spread)

L319, change (can contribute to the development or suppression of infection) into (can contribute to developing or suppressing infection)

L324, change (replication of influenza A virus) into (replication of the influenza A virus)

L327, correct (gut bcateria) into (gut bacteria)

L328, (treatment with a mic of Escherichia coli) what is meant by mic in this context? Please revise and correct.

L331, change (studying infection) into (studying the infection)

L333-334, change (Depletion of vaginal microbiome in mice was shown to increase IL-33 production) into (Depleting the vaginal microbiome in mice increased IL-33 production)

L356-357, change (to define in their role) into (to define their role)

L341, change (virus thereby)  into (virus, thereby)

L371-372, change (For instance, the gut dysfunction) into (For instance,  gut dysfunction)

L373, change (HIV infection leads to the development of neurological sympton) into (HIV infection lead to neurological symptoms)

L383-384, change (in the development of COVID-19) into (in developing COVID-19)

L386, change (can lead to the translocation of) into (can translocate)

L388, change (It is likely that interrelated mechanisms contribute to) into (Interrelated mechanisms may contribute to)

L400, change (by a variety of factors) into (by various factors)

L406-407, change (Dysbiosis can promote inflammation and oxidative stress, which are key drivers of cardiovascular disease) into (Dysbiosis can promote inflammation and oxidative stress, key cardiovascular disease drivers).

L408, change (points to a role) into (points to the role)

L418, change (TMA which) into (TMA, which)

L421, change ([129]. . Moreover) into ([129]. Moreover)

L424, change (, which is a chronic) into (, a chronic)

L441, change (promoting the accumulation of fat) into (promoting fat accumulation)

L441, change ([135] .) into ([135].)

L442, change (cytokines such as) into (cytokines, such as)

L444, change (underlying pathogenesis of nonalcoholic) into (underlying pathogenesis of nonalcoholic)

L445, change (HIV infected persons) into (HIV-infected persons)

L445, change (, which in turn contributes to higher risk) into (, contributing to a higher risk)

L448, change (Although, FLD) into (Although FLD)

L449, change (double than in) into (double that in)

L455, change (are needed for the development of novel) into (are needed to develop novel)

L456, change (measures that can prevent or slow acceleration of) into (measures to prevent or slow the acceleration of)

L464, change (illnesses such as) into (illnesses, such as)

L467, change (shown significant higher levels) into (showed significantly higher levels)

L473-474, change (and is being explored ……, including HIV and influenza virus [145] Into (It is being explored for infectious viral diseases, including HIV and influenza [145]

L475, change (of the gut permeability) into (of gut permeability)

L479, change (FMT in chickens was shown to improve resistance against avian) into (FMT in chickens improved resistance against the avian)

L483-484, change (mice that had been previously treated) into (mice previously treated)

L492, change (immune function) into (immune functions)

L494, change (Biliński et al) into (Biliński et al.)

L497, change (be conducted to clearly assess the safety of FMT.) to (be conducted to assess the safety of FMT.)

L498, change (donors including) into (donors, including)

L505, change (duration of such intervention) to (duration of such an intervention)

L508-509, change (mucosal sites that are also inhabited by commensal microbiota.) into (mucosal sites inhabited by commensal microbiota.)

L512, change (sites and thus, ensure its own survival.) into (sites, thus ensuring its own survival.)

L515-516, change (Therefore, it is important to consider) into (Therefore, it is vital to consider)

L517-518 , change (determine the outcomes of disease. ) into (determine disease outcomes.)

L522-523, change (an important immune cell subpopulation that is involved in both immune response) to (a significant immune cell subpopulation involved in both immune responses)

L524, change (Overall, gut microbiota) to (Overall, the gut microbiota)

L525, change (has great therapeutic potential.) to (has excellent  therapeutic potential.)

L527, change (it is crucial to establish) into (it is crucial to develop)

Minor linguistic modifications

L36, change (extensive and diverse) into (an extensive and diverse)

L36-37, correct (microoganisms) into (microorganisms)

L38, change (Importantly) into (Notably).

L39, change (a range of) into (various).

L40, change (various aspects of human health) into (aspects of human health)

L49, change (serves as) into (is).

L51, change (specifically targets pathogens) into (targets pathogens specifically)

L56, change (that determine the outcome) into (, determining the outcome).

L61, change (In a reciprocal manner,) into (Reciprocally,)

L67, change (such as) into (, such as)

L67-68, change (short- chain-fatty acids) into (short- chain fatty acids)

L68, change (that have) into (, that have)

L81, change (, severe acute) into (, and severe acute)

L85, change (integrity) into (integrity, )

L92, change (can contribute to the development of life-threatening) into (can contribute to developing life-threatening)

L96, change (including HIV) into (, including HIV)

L106, change (in turn) into (, in turn)

L107, change (et al,) into (et al.)

L108, change (on the activity) into (in the activity)

L115, change (, which is composed of a single) into (, composed of a single)

L118, change (including) into (, including)

L122, change (through induction) into (through the induction)

L132, change (including) into (, including)

L141, change (viruses the target) into (viruses that target)

L158, change (, which primarily targets) into (, primarily targeting)

L164, change (integrity which) into (integrity, which)

L181, change (HIV infected) into (HIV-infected)

L185, change (species that are consistent with reports in chronic) into (species consistent with reports of chronic)

L202, change (important features) into (essential features)

L202, change (progression such as) into (progression, such as)

L204, change (and increase in inflammatory) into (and an increase in inflammatory)

L209, change (respiratory viral infections) into (viral infections)

L201, change (including) into (, including)

L214, change (remains) into (remain)

L240, change (important) into (critical)

L257-258, change (could lead to disruption of this balance) into (could disrupt this balance)

L260, change (related with the severity of disease,) into (related to the severity of the disease,)

L261, change (show greater alterations) into (show more significant alterations)

L265, change (Although, the exact) into (Although the exact)

L268, change (between gut microbiome) into (between the gut microbiome).

L272-275, please revise carefully this sentence.

L275, change (several important functions) into (several essential functions)

L280, change (Extensive research over the last decade has improved the) into (Over the last decade, extensive research has improved the)

L283, change (a prominent) into (a major)

L291, change (can directly kill of infected cells) into (can directly kill infected cells)

L305, change (infections, while TLR8 activation) into (infections. At the same time, , TLR8 activation)

L311-312, change (in the detection, elimination, and control of viral spread) into (in detecting, eliminating, and controlling viral spread)

L319, change (can contribute to the development or suppression of infection) into (can contribute to developing or suppressing infection)

L324, change (replication of influenza A virus) into (replication of the influenza A virus)

L327, correct (gut bcateria) into (gut bacteria)

L328, (treatment with a mic of Escherichia coli) what is meant by mic in this context? Please revise and correct.

L331, change (studying infection) into (studying the infection)

L333-334, change (Depletion of vaginal microbiome in mice was shown to increase IL-33 production) into (Depleting the vaginal microbiome in mice increased IL-33 production)

L356-357, change (to define in their role) into (to define their role)

L341, change (virus thereby)  into (virus, thereby)

L371-372, change (For instance, the gut dysfunction) into (For instance,  gut dysfunction)

L373, change (HIV infection leads to the development of neurological sympton) into (HIV infection lead to neurological symptoms)

L383-384, change (in the development of COVID-19) into (in developing COVID-19)

L386, change (can lead to the translocation of) into (can translocate)

L388, change (It is likely that interrelated mechanisms contribute to) into (Interrelated mechanisms may contribute to)

L400, change (by a variety of factors) into (by various factors)

L406-407, change (Dysbiosis can promote inflammation and oxidative stress, which are key drivers of cardiovascular disease) into (Dysbiosis can promote inflammation and oxidative stress, key cardiovascular disease drivers).

L408, change (points to a role) into (points to the role)

L418, change (TMA which) into (TMA, which)

L421, change ([129]. . Moreover) into ([129]. Moreover)

L424, change (, which is a chronic) into (, a chronic)

L441, change (promoting the accumulation of fat) into (promoting fat accumulation)

L441, change ([135] .) into ([135].)

L442, change (cytokines such as) into (cytokines, such as)

L444, change (underlying pathogenesis of nonalcoholic) into (underlying pathogenesis of nonalcoholic)

L445, change (HIV infected persons) into (HIV-infected persons)

L445, change (, which in turn contributes to higher risk) into (, contributing to a higher risk)

L448, change (Although, FLD) into (Although FLD)

L449, change (double than in) into (double that in)

L455, change (are needed for the development of novel) into (are needed to develop novel)

L456, change (measures that can prevent or slow acceleration of) into (measures to prevent or slow the acceleration of)

L464, change (illnesses such as) into (illnesses, such as)

L467, change (shown significant higher levels) into (showed significantly higher levels)

L473-474, change (and is being explored ……, including HIV and influenza virus [145] Into (It is being explored for infectious viral diseases, including HIV and influenza [145]

L475, change (of the gut permeability) into (of gut permeability)

L479, change (FMT in chickens was shown to improve resistance against avian) into (FMT in chickens improved resistance against the avian)

L483-484, change (mice that had been previously treated) into (mice previously treated)

L492, change (immune function) into (immune functions)

L494, change (Biliński et al) into (Biliński et al.)

L497, change (be conducted to clearly assess the safety of FMT.) to (be conducted to assess the safety of FMT.)

L498, change (donors including) into (donors, including)

L505, change (duration of such intervention) to (duration of such an intervention)

L508-509, change (mucosal sites that are also inhabited by commensal microbiota.) into (mucosal sites inhabited by commensal microbiota.)

L512, change (sites and thus, ensure its own survival.) into (sites, thus ensuring its own survival.)

L515-516, change (Therefore, it is important to consider) into (Therefore, it is vital to consider)

L517-518 , change (determine the outcomes of disease. ) into (determine disease outcomes.)

L522-523, change (an important immune cell subpopulation that is involved in both immune response) to (a significant immune cell subpopulation involved in both immune responses)

L524, change (Overall, gut microbiota) to (Overall, the gut microbiota)

L525, change (has great therapeutic potential.) to (has excellent  therapeutic potential.)

L527, change (it is crucial to establish) into (it is crucial to develop)

Author Response

We thank the reviewer for this comprehensive review with comments and suggestions for improving our manuscript. We have made the following changes:

Article structure:

We have modified the sequential structure of the sub-sections based on the reviewer’s suggestion. 

Figure:

We have now included all the abbreviations appearing in the figure directly after the title.

The text from L145 to L155 is now separated from the title and be presented as an independent paragraph.

We have included all the minor linguistic modifications suggested as highlighted text.